# Insights into Pre-training via Simpler Synthetic Tasks

**Yuhuai Wu**[12*]
yuhuai@cs.stanford.edu

**Felix Li**[3*]
fzli@berkeley.edu

**Percy Liang**[1]
pliang@cs.stanford.edu

[1]Stanford University
[2]Google Research
[3]UC Berkeley

## Abstract

Pre-training produces representations that are effective for a wide range of downstream tasks, but it is still unclear what properties of pre-training are necessary for effective gains. Notably, recent work shows that even pre-training on synthetic tasks can achieve significant gains in downstream tasks. In this work, we perform three experiments that iteratively simplify pre-training and show that the simplifications still retain much of its gains. First, building on prior work, we perform a systematic evaluation of three existing synthetic pre-training methods on six downstream tasks. We find the best synthetic pre-training method, `LIME`, attains an average of $67\%$ of the benefits of natural pre-training. Second, to our surprise, we find that pre-training on a simple and generic synthetic task defined by the `Set` function achieves $65\%$ of the benefits, almost matching `LIME`. Third, we find that $39\%$ of the benefits can be attained by using merely the parameter statistics of synthetic pre-training. We release the source code at https://github.com/felixzli/synthetic_pretraining.

## 1 Introduction

Pre-training on a large amount of data from sources such as text from the web—*natural pre-training*—is effective for a wide range of downstream tasks (Devlin et al., 2019; Brown et al., 2020; Chen et al., 2021; Lu et al., 2021). More surprisingly, recent works (Wu et al., 2021; Krishna et al., 2021; Ri & Tsuruoka, 2022; Anderson & Farrell, 2022) show that pre-training on data that is fully synthetically generated—*synthetic pre-training*—can also provide substantial gains, partially closing the gap between training from a randomly initialized model and a naturally pre-trained model.

What properties of pre-training are necessary for effective gains? In this paper, we conduct a careful empirical study to help understand this question. We perform three experiments that iteratively simplify natural pre-training, while still retaining much of the benefits of natural pre-training. See Figure 1 for an overview of the results obtained in our three experiments.

First, we perform a systematic evaluation of three previously proposed synthetic pre-training methods over six downstream tasks. Prior works only evaluated each synthetic pre-training method on a single downstream domain: Wu et al. (2021) on mathematical reasoning benchmarks, Chiang & Lee (2021) on GLUE, Krishna et al. (2021) on summarization, and Ri & Tsuruoka (2022) on language modeling and dependency parsing. We instead show how general these methods are as well as how different synthetic pre-training methods compare against each other. We find that the best performing synthetic task, `LIME` (Wu et al., 2021), closes a significant fraction of the gap between random initialization and natural pre-training for summarization ($85\%$), semantic parsing ($83\%$), reading comprehension ($55\%$), code translation ($43\%$), and retrosynthesis ($49\%$).

---

[*]Equal Contribution.

36th Conference on Neural Information Processing Systems (NeurIPS 2022).

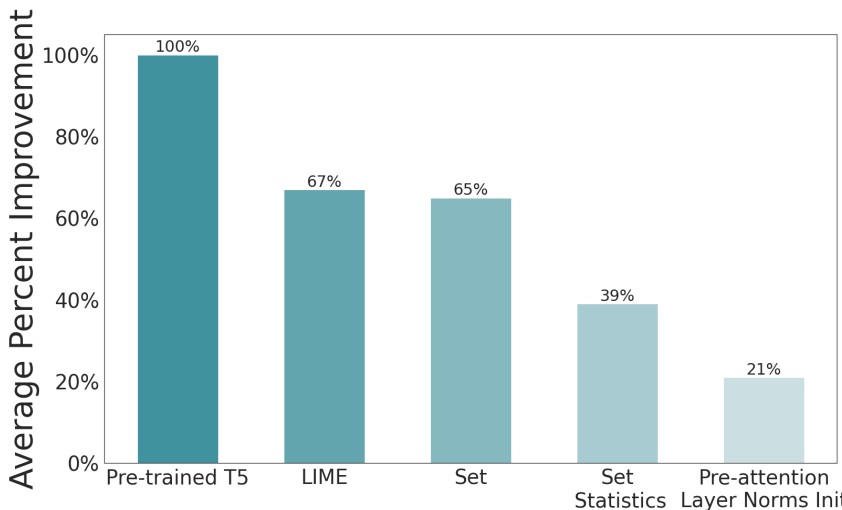

Figure 1: We compare downstream task performance between natural language pre-training (`Pre-trained_T5`), synthetic pre-training (`LIME`), simpler synthetic pre-training (`Set`), initializing from the statistics of Set pre-training (`Set` Statistics), and initializing just the pre-attention layer norms to a lower value (Pre-attention Layer Norm Init). The average percent improvement is computed by taking the average over our six downstream tasks of $\frac{\text{Performance(Initialization)} - \text{Performance(Random\_Init)}}{\text{Performance(Pre-trained\_T5)} - \text{Performance(Random\_Init)}}$.

Second, we discover a surprisingly simple and generic synthetic task that captured nearly the same effectiveness provided by the existing best synthetic task `LIME`, closing an average of $65\%$ of the gap to natural pre-training (compared to $67\%$ for `LIME`). This simple task, which we call `Set`, consists of outputting the input sequence without duplicate tokens in its original order, for example:

$$\text{Input: a a c b a b} \Rightarrow \text{Output: a c b.}$$

Third, to simplify the initialization even more, we ask whether only the statistics of the pre-trained weights can explain the effectiveness of synthetic pre-training. We find that initializing parameters with statistics of a synthetic pre-trained model provides large benefits for some downstream tasks, closing $39\%$ of the gap to natural pre-training. Simplifying further allows us to see that by initializing all pre-attention layer norm parameters to a lower value (just one number!), we close more than $40\%$ of the gap to natural pre-training performance for summarization and semantic parsing, and an average of $21\%$ over all six tasks.

## 2   Experimental Setup

**Architecture**   We trained a T5-Small model, which is a 60 million parameter encoder-decoder transformer model with 6 encoder layers, 6 decoder layers, 8 heads, head dimension 64, and MLP dimension 2048 (Raffel et al., 2020).

**Training Details**   For synthetic pre-training, we use the same hyperparameters that the off-the-shelf language pre-trained T5-small was trained with: AdaFactor optimizer, batch size 128, sequence length 512, and inverse square root learning rate $1/\sqrt{\max(n, 10000)}$ where $n$ is the current training step. We evaluate token validation accuracy every 5000 training steps. For all synthetic tasks besides `Dyck`, we fine-tuned with the first checkpoint that the model reaches above 99% token validation accuracy. For `Dyck`, the model's validation accuracy plateaued without reaching above 99%, so we chose to fine-tune using the checkpoint with max validation accuracy of 77.7%. We provide further pre-training details in Appendix A and fine-tuning details in Appendix B.

**Downstream Tasks**   We fine-tuned synthetically pre-trained models on a diverse suite of downstream tasks: 1) Java to C# code translation (10K training examples) (Lu et al., 2021); 2) two semantic parsing benchmarks, MTOP (17K training examples) (Li et al., 2021) and WebQSP (2.7K training

Table 1: Evaluation of three previously proposed synthetic pre-training tasks and two simpler synthetic pre-training tasks. For tasks where we report both Exact Match (EM) and F1, the Average column is computed using EM.

| | CNNDM-10K ROUGE1 | MTOP EM/F1 | WebQSP EM/F1 | SQuAD EM/F1 | Code Trans. EM | Retrosyn. EM | Average | |
|---|---|---|---|---|---|---|---|---|
| | | | | Baselines | | | | |
| Pre-trained_T5 | **35.8**[1] | **81.0/95.2** | **83.1/91.9** | **77.5/86.0** | **61.6** | **43.1** | **63.7** | **100%** |
| Random_Init | 18.9 | 38.6/83.0 | 28.2/72.9 | 17.2/25.2 | 57.0 | 39.2 | 33.2 | **0%** |
| | | | Section 3: Benchmarking Existing Synthetic Pre-training | | | | | |
| LIME | **33.2** | **73.7/94.0** | **75.2/88.7** | **50.4/62.1** | **59.0** | **41.1** | **55.4** | **67%** |
| Dyck | 27.1 | 65.9/91.9 | 58.5/83.5 | 50.3/62.4 | 58.8 | 40.4 | 50.2 | 49% |
| Nons_Summary[2] | 32.0 | 68.0/92.7 | 65.2/85.2 | 48.4/60.1 | 57.3 | 39.6 | 51.8 | 47% |
| | | | Section 4: Simpler Synthetic Pre-Training | | | | | |
| Set | **32.8** | **71.7/93.7** | **72.2/88.2** | **48.3/60.0** | **59.4** | **40.9** | **54.2** | **65%** |
| Identity | 30.2 | 68.6/93.0 | 69.8/86.8 | 26.1/35.4 | 57.8 | 40.5 | 48.8 | 46% |

examples) (Yih et al., 2016), consisting of converting a natural language query to a logical form; 3) USPTO-50K retrosynthesis (40K training examples) (Liu et al., 2017), a task that consists of predicting possible reactants when given a product as input; 4) the reading comprehension benchmark SQuAD 1.1 (87K training examples) (Rajpurkar et al., 2016); and 5) the summarization benchmark CNNDM-10K[3] which is 10K training examples from the CNNDM (Krishna et al., 2021). See full descriptions and input-output examples for every downstream task in Appendix D.

**Baselines**  We compared against randomly initialized (Random_Init) and off-the-shelf natural language pre-trained T5-Small (Pre-trained_T5). Pre-trained_T5 was trained for 524K steps on the C4 dataset with the span-corruption objective (Raffel et al., 2020). We also compared against T5-Small trained on Wikipedia (Guo et al., 2020) for 10K steps (Wiki_10K) with the same pre-training objective as Pre-trained_T5.

**Source code**  We release the source code to reproduce the experiments at https://github.com/felixzli/synthetic_pretraining.

## 3 Benchmarking Existing Synthetic Pre-training

We benchmark three previously proposed synthetic pre-training tasks across six downstream tasks. We compute how much of the benefits using natural pre-training are gained by each synthetic pre-training method by taking the average over six downstream tasks of $\frac{\text{Performance(Initialization)} - \text{Performance(Random\_Init)}}{\text{Performance(Pre-trained\_T5)} - \text{Performance(Random\_Init)}}$. We show these numbers in the right-most column of Table 1. We find that LIME achieves the best performance, gaining 67% of the benefits using natural pre-training.

### 3.1 Synthetic Tasks

For each synthetic task, we generated one million examples. To generate synthetic task data, we sampled tokens a vocabulary of size 32K, matching the vocabulary size of Pre-trained_T5. Details about data generation can be found in the original papers. We provide an example of each task in Figure 2.

---

[1]This number was obtained using pre-trained T5_1.1 instead of T5_1.0 which we used for all other tasks. Using pre-trained T5_1.0 would be an unfair comparison because it has already been trained on all 290K CNNDM task training data. Full differences between T5_1.1 and T5_1.0 are described in Appendix Section C.

[2]We obtained different results than Krishna et al. (2021) on CNNDM-10K because of our choice of optimizer. We were able to match their results when we used Adam like they did in their work instead of Adafactor. We chose to use Adafactor instead of Adam because training took about three times less steps.

[3]We followed Krishna et al. (2021) to evaluate on CNNDM-10K instead of the full CNNDM. They used CNNDM-10K because less training examples makes the impact of pre-training more visible.

**LIME Deduct**

Input:
```
<RuleSymbols> ABC
<MathSymbols> *+=abde
 A*A+B=C
 {A:a,B:b,C:d+e}
```

Output:
```
a*a+b=d+e
```

**Dyck**

Input:
```
[ ▢ ▢ [ ] } ▢ ] )
 ▢ { [ ] ▢
```

Output:
```
(  {  [
 ]  }
```

**Nonsense Summary**

Input:
```
dkb spf hpd vfb nwg.
uwa lja oyg mjg ige lqc
rbb oeh d10_keyword_1
vwg zob jdf quc .
d3_keyword_7 vgc bge
epf slb ecd .
```

Output:
```
lqc rbb oeh
d10_keyword_1
vwg zob jdf quc .
the keyword was
negative .
```

**Set**

Input:
```
a a c b a b
```

Output:
```
a c b
```

**Identity**

Input:
```
a a c b a b
```

Output:
```
a a c b a b
```

Figure 2: We show examples of Section 3 previously proposed synthetic tasks in the top row and Section 4 simpler synthetic tasks in the bottom row. **Top Left:** LIME deduct task, whose input consists of a "rule string" and "substitution dictionary" and output is a "result string." The "rule symbols" and "math symbols" are written as actual letters and math operations for reader clarity. In real generated data, each example's rule and math symbols are randomly sampled tokens. **Top Middle:** T5 style masked language modelling with Dyck language. In the input, the grey boxes indicate masked out spans. The model is trained to predict the masked spans, each separated by a special token . In this example, the "noise_density" is 0.33 (so 5 out of 15 input brackets are corrupted) and the "span_corruption_length" is 1 (so each masked span is of length 1). In actual generated data, the "span_corruption_length" is 3 and the "noise_density" is 0.15, which are the same parameter values used for natural language Pre-trained_T5. **Top Right:** Nonsense_Summary task. In this example, two operations to create the summary are "copy sentence containing a keyword" and "identify keyword sentiment." **Bottom Left:** Set task. **Bottom Right:** Identity task.

**LIME** LIME is a set of three tasks—Deduct, Induct, and Abduct—inspired by Charles Peirce's three reasoning primitives (Wu et al., 2021). Each task consists of three elements: a rule string, a dictionary that represents substitutions, and a result string that is the result of applying those substitutions to the rule string. The three tasks are then constructed by using two of the three elements as inputs to predict the remaining element.

**Dyck Artificial Language** We consider a Dyck language with multiple bracket types (taken from Ri & Tsuruoka (2022)), which we refer as Dyck. We turn generated text into a sequence-to-sequence task by using the T5 span-corruption pre-training objective (Raffel et al., 2020). The motivation behind the design of Dyck is how "sentences of natural language often have dependency relations where the existence of a certain word can be predictive of another word" (Ri & Tsuruoka, 2022).

**Nonsense Summary** Krishna et al. (2021) developed the nonsense summarization task, which we refer to as Nonsense_Summary, by conducting a qualitative analysis of summarization data and identifying 21 elementary operations used to create summaries such as "CopyFirstSentence" and "CopyQuoted." Each synthetic task example consists of an input that is a nonsense document made of randomly sampled tokens and an output that is the corresponding nonsense summary consisting of applying a random subset of the 21 operations to the document.

### 3.2 Results

Table 1 shows results of evaluating the synthetic tasks across six downstream tasks, and Appendix Table 4 shows results of ablating the dataset size for retrosynthesis, CNNDM, and SQuAD.

**Synthetic pre-training provides large gains over random initialization.** We observed large gains from all three synthetic tasks we evaluated. Out of these, LIME provided the most benefits across all six downstream tasks. We computed how much of the natural pre-training benefit is captured by LIME as $\frac{\text{Performance(LIME)} - \text{Performance(Random\_Init)}}{\text{Performance(Pre-trained\_T5)} - \text{Performance(Random\_Init)}}$. LIME closed $84.6\%$, $82.8\%$, $85.6\%$, $55.1\%$, $43.5\%$, and $48.7\%$ of the gap for CNNDM-10K, MTOP, WebQSP, SQuAD, Code Translation, and Retrosynthesis respectively.

**Task or domain prior is not required for pre-training benefits.** For example, LIME benefits tasks including summarization (CNNDM-10K) and reading comprehension (SQuAD) despite LIME pre-training data involving no natural language like Pre-trained_T5, representing no structure of natural language like Dyck, and not being explicitly designed to reflect the operations used in summarization like Nonsense_Summary.

**Varying degrees of benefits.** Comparing Nonsense_Summary and Dyck, we see that the two synthetic tasks benefit MTOP, WebQSP, and SQuAD similarly, yet their benefits for the three other tasks are very different. The difference between Nonsense_Summary and Dyck as well as the superiority of LIME are examples of how different synthetic pre-training can have significantly different effectiveness.

**Synthetic pre-training is not more efficient.** One of the advantages argued for synthetic pre-training is its computational efficiency (Wu et al., 2021; Krishna et al., 2021). To explore this potential advantage, we pre-trained our model using the T5 span-corruption pre-training objective on Wikipedia (Guo et al., 2020) for 10K steps, and the result is shown in Table 1 labelled Wiki_10K. Wikipedia pre-training for 10K steps provided more significant gains than LIME pre-training for 30K steps and Set pre-training for 10K steps. Based on this result, we see no evidence that the evaluated synthetic pre-training tasks provides computational efficiency benefits over natural language pre-training, unlike what Wu et al. (2021) suggested.

**Natural language pre-training benefits non-natural language tasks.** Natural language pre-training, including Wiki_10K, outperforms synthetic pre-training on the non-natural language tasks of Code Translation and Retrosynthesis.

## 4 Simpler Synthetic Pre-training

Encouraged by the results in the previous section, we want to explore what is necessary for synthetic pre-training benefits. We evaluate pre-training with simpler and more generic synthetic tasks similar to basic Python API functions. Surprisingly, a simple and generic task defined by the Set function provides significant benefits ($65\%$), almost matching the previous best synthetic task LIME ($67\%$). Furthermore, another simple task defined by the Identity function also provided significant benefits ($46\%$).

### 4.1 Simpler Synthetic Tasks

We constructed 18 simpler synthetic tasks similar to basic Python API functions. As we did for the three existing synthetic tasks, for each simpler synthetic task we generated one million data points with a vocabulary of 32K tokens, matching the vocabulary size of Pre-trained_T5. In Appendix G.1, we define each task, explain how examples are randomly generated, and provide input-output examples. We first evaluated all tasks on CNNDM-10K. We then evaluated Set, one of the best performing tasks on CNNDM-10K, and Identity, one of the simplest tasks, on all six downstream tasks.

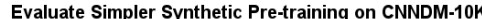

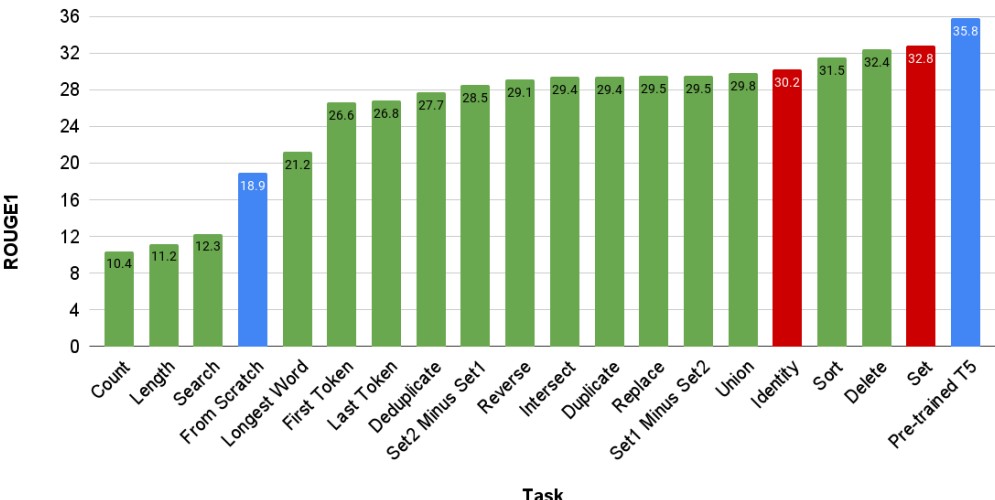

Figure 3: For each simple synthetic task, we pre-train on it and then fine-tune on CNNDM-10K. `Set` performed the best and `Identity`, one of the simplest tasks, provided significant benefit. The `Set` and `Identity` tasks are defined in 4.1, and the other tasks are defined in G.

**Set**   The `Set` task consists of outputting the input sequence in its original order except without tokens appearing more than once. An example input is [a b b a a a c d c], and the corresponding output is [a b c d]. The `Set` task data generation is explained in Appendix G.

**Identity**   The `Identity` task consists of copying the input tokens to the output. An example input is [a d e a a], and the corresponding output is [a d e a a]. To generate one example, we uniformly sample a sequence of tokens between length 10 and length 220 to be the input and output.

## 4.2   Results: simpler synthetic tasks provide similar benefits as existing synthetic tasks

Figure 3 shows results on CNNDM-10K for models pre-trained on each of 18 simpler synthetic tasks. Table 1 shows results on all six downstream tasks for `Set` and `Identity` pre-trained models. Averaged across the six downstream tasks, `Set` and `Identity` gain 65% and 46% of the natural pre-training benefits compared to 67% for `LIME`, the best previously proposed synthetic task evaluated in Section 3. `Identity` provides at least as much benefit as Nested Language and Nonsense Summary for all tasks besides Code Translation, for which Nested Language benefits more.

The significant benefits of `Set` and `Identity` pre-training suggest that the complexities of the three previously proposed synthetic tasks we evaluated—representing three fundamental logical operations, having a structural property that mimics natural language, and reflecting operations used for summarization—may not be the necessary factors for why pre-training on those tasks provides benefits. Also, due to their simplicity, `Set` and `Identity` may be useful for future work towards understanding pre-training and how to design better synthetic tasks.

## 5   Even Simpler Synthetic Pre-training with Statistics

Inspired by the previous section, we tried to find even simpler initializations that achieve the the benefits of pre-training. We explored how benefits of synthetic pre-training can be attained by initialization based only on statistics of pre-trained weights. We found: (1) Initialization using the mean and standard deviation (SD) of synthetic pre-trained parameters can gain more than 70% of the benefit from synthetic pre-training for CNNDM-10K and MTOP (or an average of 39% of natural pre-training benefits). (2) Initialization using simpler schemes involving less statistics can still provide benefits. (3) Specifically the pre-attention layer norm plays an important role for initialization benefits.

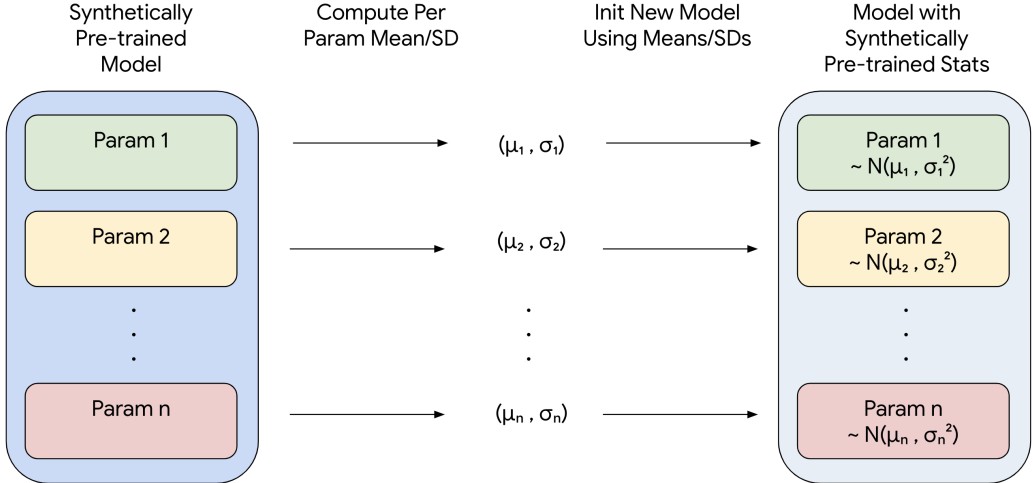

Figure 4: Illustration of Section 5.1 `Per_Param_Mean_SD` init scheme. We use this init scheme with `LIME`, `Set`, or `Identity` pre-trained models before fine-tuning on the downstream tasks. As an example, if we were to use this init scheme with an `Identity` pre-trained model, we would initialize the encoder layer 2 query matrix weights by sampling from a Gaussian with the the mean and SD of the encoder layer 2 query matrix from an `Identity` pre-trained model, and we would initialize every other parameter in the same manner. The `Per_Param_Mean_SD` init scheme recovered more than 70% of the benefit of normally fine-tuning a synthetic pre-trained model (or an average of 39% of natural pre-training benefits).

## 5.1 How much of synthetic pre-training benefit can be attributed to better statistics for initialization?

We compare normal fine-tuning, which we call `Full_Init`, with fine-tuning a model that is initialized with the mean and standard deviation (SD) of each synthetic pre-trained model's parameter. We call this initialization scheme `Per_Param_Mean_SD` and illustrate it in Figure 4. Specifically, for each parameter $p$ (e.g. an attention query matrix) in the synthetic pre-trained model, we compute the the mean $\mu_p$ and the standard deviation $\sigma_p$ of all entries in $p$. We then randomly initialize the model with the same $\mu_p$ and $\sigma_p$ for each parameter $p$. For example, if a pre-trained model had a $2 \times 2$ parameter matrix $p = [[-1, 2], [2, 5]]$, then $\mu_p = 2$ and $\sigma_p = 2.4$, and we would initialize $p$ by sampling the each entry of this $2 \times 2$ matrix from a normal distribution $\mathcal{N}(2, 2.4^2)$. We list all all the parameters we initialize for T5-Small in this manner in Appendix K.

As shown in Table 2, using the `Per_Param_Mean_SD` initialization scheme achieves a large proportion of the benefit from normally fine-tuning a synthetic pre-trained model. For example, for `LIME` `Identity` and `Set`, the `Per_Param_Mean_SD` initialization recovered 78%, 85%, and 97%[2] of MTOP `Full_Init` performance (or 65%, 66%, 69% of the benefits by natural pre-training). The remaining gap between `Per_Param_Mean_SD` and `Full_Init` supports the possibility that other factors besides better initialization statistics may cause the benefits of synthetic pre-training. We note that it does not preclude the possibility that better initialization statistics is the sole cause of benefits: there may exist other initialization schemes such that the statistics of the pre-trained model are a subset of the statistics from the initialization scheme.

## 5.2 Can simpler initialization schemes still produce benefits?

Next we explore if simpler initialization schemes that use a smaller number of statistics than `Per_Param_Mean_SD` can still provide benefits. We present three initialization schemes: `Per_Param_Scale`, `Whole_Model_Scale`, and `Subset_Per_Param_Scale`, with results shown in Table 2. We also tried two other initialization schemes, `Across_Layers_Scale` and `Per_Layer_Scale`, discussed in Appendix H.

---

[2]These percentages are computed by $\frac{\text{performance}(\texttt{Per\_Param\_Mean\_SD}) - \text{performance}(\texttt{Random\_Init})}{\text{performance}(\texttt{Full\_Init}) - \text{performance}(\texttt{Random\_Init})}$

Table 2: Evaluating different initialization schemes with statistics of `LIME`, `Set`, and `Identity` pre-trained models. Results for `Per_Param_Mean_SD` for the four other tasks as well as five seeds for CNNDM-10K, MTOP, and SQuAD are in Appendix Table 7. Results for two other initialization ablations and three other `Subset_Per_Param_Scale` subset choices we tried are in Appendix Table 5.

| | CNNDM-10K | | | MTOP | | |
|---|---|---|---|---|---|---|
| | LIME | Set | Identity | LIME | Set | Identity |
| Section 5.1 | | | | | | |
| Full_Init | 33.2 | 32.8 | 30.2 | 73.7/94.0 | 71.7/93.7 | 68.6/93.0 |
| Per_Param_Mean_SD | 29.3 | 29.8 | 28.8 | 66.1/92.3 | 66.6/92.3 | 67.8/92.6 |
| Section 5.2 | | | | | | |
| Per_Param_Scale | 29.0 | 29.5 | 29.0 | 67.3/92.5 | 66.5/92.4 | 66.3/92.3 |
| Whole_Model_Scale | 15.5 | 26.1 | 20.5 | 34.3/81.4 | 38.7/83.1 | 59.8/91.0 |
| Pre-attn_LN_Per_Param_Scale | 27.8 | 25.1 | 27.3 | 59.9/90.3 | 58.9/89.0 | 46.7/86.3 |

| Baseline | CNNDM-10K | MTOP |
|---|---|---|
| Random_Init | 18.9 | 38.6/83.0 |
| Pre-trained_T5 | 35.8 | 81.0/95.2 |

**Per Param Scale**   We use the term "scale statistic" to denote the mean of layer norm parameters or the SD of non layer norm parameters. Unlike the `Per_Param_Mean_SD` init scheme in the previous subsection, where we initialize each parameter with two statistics (mean and SD), `Per_Param_Scale` initializes each parameter with only the scale statistic: initialize layer norm parameters by setting their value to the mean of the corresponding pre-trained parameter weights, and initialize non layer norm parameters by sampling from a Gaussian with mean 0 and the SD of the corresponding pre-trained parameter weights.

**Whole Model Scale**   We initialize the whole model with only two statistics: the SD over all non layer norm weights $\sigma_m$ and the mean over all layer norm weights $\mu_m$. Initialize each non layer norm weight by sampling from $\mathcal{N}(0, \sigma_m^2)$ and Initialize each layer norm weight as the value $\mu_m$.

**Per Param Scale: Subset**   The motivation for this init scheme is to find a smaller set of parameters whose initialization can provide significant benefit. We initialize a subset of the parameters (e.g., attention parameters) with the `Per_Param_Scale` initialization and initialize the rest of the parameters with the default `Random_Init`. With this initialization scheme, we tried four different subsets: attention parameters, MLP parameters, pre-attention layer norms, and pre-MLP layer norms.

**Results**   `Per_Param_Scale` provided almost identical benefits as the `Per_Param_Mean_SD` initialization, suggesting that just the scale statistics, rather than both the mean and SD, are sufficient to provide benefits of initialization.

`Whole_Model_Scale` initialization was much worse than `Per_Param_Mean_SD`, yielding almost no improvements. A finer scale initialization rather than simply increasing the scale of weights across the whole model as `Whole_Model_Scale` does may be necessary for benefits.

For the `Subset_Per_Param_Scale` init scheme, pre-attention layer norms was the only subset we tried that provided significant initialization benefits, covering $46\%$ and $39\%$ of the gap between From Scratch and `Pre-trained_T5` for CNNDM-10K and MTOP. See results in Table 2 (labeled `Pre-attn_LN_Per_Param_Scale`) and results for all four subsets tried in Appendix Table 5.

### 5.3 A Better Pre-attention Layer Norm Initialization

**Synthetic pre-training results in lower pre-attention layer norm means.**   Motivated by benefits from the `Subset_Per_Param_Scale` initialization scheme—initializing pre-attention layer norms using their synthetic pre-trained means and all other parameters with the default initialization—we wanted to see if the pre-attention layer norm mean values in synthetic pre-trained models have a

Table 3: Results from initializing pre-attention layer norm parameters to a particular value. They are initialized to the value of 1.0 in the `Random_Init` baseline which is the default T5 initialization.

| Pre-attn LN Init Value | CNNDM-10K | MTOP | WebQSP | SQuAD | Code Trans. | Retrosyn. |
|---|---|---|---|---|---|---|
| 0.05 | 28.2 | 25.7/73.3 | 30.5/71.2 | 22.8/31.2 | 57.9 | 39.3 |
| 0.1 | 28.3 | 32.5/77.5 | 28.9/70.6 | 23.5/31.7 | 57.6 | 39.9 |
| 0.2 | 28.2 | 56.4/89.3 | 30.6/72.3 | 28.2/37.3 | 57.0 | 39.4 |
| 0.4 | 24.4 | 56.9/89.9 | 30.0/72.8 | 36.3/46.9 | 57.3 | 39.6 |
| 0.8 | 18.6 | 50.7/88.5 | 28.0/71.6 | 34.2/44.6 | 57.2 | 39.9 |
| | | | | | | |
| Baseline | CNNDM-10K | MTOP | WebQSP | SQuAD | Code Trans. | Retrosyn. |
| `Random_Init` | 18.9 (0.5) | 38.6/83.0 (2.1/1.2) | 25.8/71.5 (0.7/0.5) | 16.0/24.2 (2.1/2.0) | 57.6 (0.3) | 39.0 (0.5) |
| `Pre-trained_T5` | 35.8 (0.1) | 81.0/95.2 (0.3/2.2) | 82.3/91.7 (0.8/0.4) | 77.4/86.1 (0.2/0.2) | 61.2 (0.7) | 43.5 (0.3) |

pattern. In Appendix Figure 6, we plotted pre-attention layer norm parameter means of each layer for `LIME`, `Set`, and `Identity` pre-trained models. We noticed that the pre-attention layer norm means from the synthetic pre-trained models are much smaller values (below 0.4 for most layers) compared to the default initialization of 1.0.

**Lower value pre-attention layer norm initialization provides benefits.** Inspired by the above observation, we tried an initialization scheme where all the pre-attention layer norm parameters are initialized to a specific value, and all other parameters are initialized with the default initialization. We tried five different values ranging from 0.05 to 0.8. Results in Table 3 show that initializing the pre-attention layer norm parameters to a lower value than the default initialization of 1.0 provided significant benefits: initializing them to 0.2 improved CNNDM-10K from 18.9 to 28.2 ROUGE1, MTOP from 38.6% to 56.4% EM, and SQuAD from 16.0 to 22.8 EM.

## 6 Related Work

**Synthetic Pre-training** Past work has shown that synthetic pre-training can benefit a wide range of downstream tasks. For example, it has been shown that pre-training on artificial languages that intuitively mimic properties of natural language can benefit language modeling, dependency parsing, and GLUE (Ri & Tsuruoka, 2022; Papadimitriou & Jurafsky, 2020; Chiang & Lee, 2021). Pre-training on LIME, a mixture of three synthetic tasks motivated from reasoning primitives, benefits mathematical reasoning benchmarks (Wu et al., 2021). For summarization, Krishna et al. (2021) designed a suite of synthetic summarization tasks on a corpus of random tokens, and they show such pre-training can almost match up with natural pre-training. Synthetic pre-training also has been applied outside of text-based domains: Anderson & Farrell (2022) shows pre-training on synthetically generated fractal images nearly matches the benefits of pre-training with ImageNet.

**Cross-domain Transfer Learning** Synthetic pre-training benefiting downstream tasks across a wide range of domains is an instance of cross-domain transfer learning, which past work has demonstrated the efficacy of. Pre-training language models with many languages benefits downstream tasks in a new unseen language (Liu et al., 2020). More surprisingly, pre-training on music, code, and amino acid sequences benefits natural language downstream tasks (Papadimitriou & Jurafsky, 2020; Chiang & Lee, 2020). It was also shown recently that pre-training on natural language benefits offline RL (Reid et al., 2022).

**Understanding Pre-training** Past work has explored what properties of the pre-training data are important for benefits. Ri & Tsuruoka (2022) and Papadimitriou & Jurafsky (2020) use synthetic pre-training to identify general properties of pre-training data that correlate with improved downstream language task performance by leveraging how synthetic data is easy to modify in a controlled manner. Papadimitriou & Jurafsky (2020) found that changing the distribution of tokens in an artificial language can affect downstream performance, and Ri & Tsuruoka (2022) found that for an artificial brackets language using a nesting structure as well as different tokens for bracket pairs results in better performance than using a flat structure and the same token for bracket pairs. K et al. (2020), Conneau et al. (2020), and Dufter & Schütze (2020) explore data properties related to cross-language transfer learning. Sinha et al. (2021) shows that word order within sentences is not essential for natural langauge pre-training benefits.

Other past work has used probing methods to explore what potentially transferable knowledge is gained from pre-training. Ri & Tsuruoka (2022) explored how much knowledge of "position-aware context dependence" was gained from artificial language and natural language pre-training. Chi et al. (2020) and Papadimitriou et al. (2021) explored potentially transferable knowledge gained from multi-lingual language pre-training.

**Initialization** Our work tried to understand pre-training through the lens of initialization statistics that benefit optimization. There has been past work on improving transformer initialization to achieve more stable training and benefits with deeper models (Zhang et al., 2019; Huang et al., 2020; Zhang et al., 2020; Wang et al., 2022). These papers do not explore benefits from initializing with pre-trained model statistics.

## 7 Discussion

**Limitations of current synthetic tasks** In this work, we show that synthetic pre-training can provide significant benefits across many different tasks. However, synthetic pre-training still lags behind natural pre-training by an average of 33% for the six downstream tasks we fine-tuned on. This is expected for natural language downstream tasks, but it is important to note that even on non-natural language tasks, natural language pre-training still outperforms synthetic pre-training. This result suggests that natural language pre-training produces effects that are missing from the evaluated synthetic pre-training. Exploring the differences between different pre-training may be an interesting direction for future work.

We emphasize that this paper does not support any claims regarding the practical efficacy of synthetic pre-training. The metric of "percentage of natural pre-training gap closed" measures relative performance along a meaningless linear scale, and for downstream tasks retrosynthesis and code translation, the magnitude of the improvement over from scratch for natural pre-training is low to begin with. This paper's goal is not to communicate that synthetic pre-training is practically effective, but rather to communicate that synthetic pre-training produces nontrivial differences in model behavior compared to a randomly initialized model, and also that there is nontrivial variation in model behavior from differing synthetic pre-training and initialization schemes involving pre-trained weight statistics.

**Synthetic pre-training requires deeper understanding** Our results show that initialization using the statistics from synthetic pre-trained models can recover more than 70% of the benefit from synthetic pre-training. The remaining gap leaves open the possibility that the benefits of synthetic pre-training are due to other factors besides better statistics for initialization. Due to their simplicity, `Set` and `Identity` may be useful for future work towards deeper understanding of synthetic pre-training.

In the long term, as we learn more about synthetic pre-training and pre-training in general, we believe that for some downstream tasks it may be possible to develop synthetic pre-training that outperforms natural pre-training: the complexity of existing natural data is fixed, while in some sense the complexity of fully synthetically generated data is infinite.

**Privacy and Ethics** This work focuses on the understanding of synthetic pre-training. But synthetic pre-training can also have practical implications for privacy and ethics. Standard pre-training data, even when its public, can contain private user information. Large pre-trained models are capable of memorizing training examples, which makes them vulnerable to privacy attacks (Carlini et al., 2020). Synthetic data on the other hand is divorced from the real world, which can help mitigate privacy issues. Large pre-trained corpora can contain harmful text (e.g., hate speech). With synthetic pre-training, since we have complete control over what goes into the data, we can potentially mitigate some of the harmful behavior of existing models. Unfortunately, synthetic pre-training is currently not as performant as natural pre-training, so real-world data would still have to be used to close the performance gap.

## Acknowledgements

We thank Google TPU Research Cloud for the experimental support. We thank Christian Szegedy, Eric Zelikman, Tri Dao, Dan Fu, Sidd Karamcheti and Rishi Bommasani, for their useful feedbacks on the draft.

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
