# OpenReview forum: "Insights into Pre-training via Simpler Synthetic Tasks"
_NeurIPS.cc/2022/Conference — NeurIPS 2022 Accept_

### Official Review · Reviewer_6Qmz · 2022-07-03

**Rating:** 4
**Confidence:** 5
**Soundness:** 2 fair
**Presentation:** 3 good
**Contribution:** 2 fair

**Summary:**

This paper presents a new insight into synthetic pre-training rather than natural language pre-training. Compared to the previous work, the paper makes experiments on a wider range of tasks (e.g., semantic parsing, code translation, natural language MRC), present more (and simpler) synthetic pre-training tasks, and explore in depth the source of gain of synthetic pre-training.
This work conducts a comprehension evaluation of synthetic pre-training methods. The key findings are three sides:
1.	Using synthetic tasks for pre-training achieves large gains compared with random initialization.
2.	Using simpler synthetic tasks like Set and Identity obtains similar benefits to existing more complex synthetic tasks.
3.	The benefits of synthetic pre-training can be attained by initializing based only on statistics.


**Questions:**

see the weakness/detailed concern 7 above.

**Ethics Review Area:**

["I don’t know"]

**Limitations:**

not see

**Strengths And Weaknesses:**

Strengths

It is an important direction to interpret the effects of pre-training methods. This work presents some interesting attempts to study the influence of different kinds of synthetic pre-training tasks that are generic vs. complex and in non-natural language vs. in natural language. It is also good to see that both generic and non-natural language tasks bring comparable gains compared with the existing methods. The findings may benefit future studies on designing better pre-training tasks.

Weakness


[Organization]

Although this work has the merits of comprehensive analysis and interesting findings, this work is not well organized. The motivation is not clear. I would suggest motivating this work by discussing some limitations of the current synthetic pre-training methods. According to the discussions in Sections 4-5, existing synthetic tasks might have disadvantages, e.g., computation complexity/cost, so they are not optimal as the pre-training tasks, and more generic ones (proposed in this work) would be better.

Besides, this paper does not give a clear answer to the core question raised in the introduction, “What properties of pre-training are sufficient for effective gains”. It would be better to clarify it in conclusions.

The experiments are presented without a principle line. The results are listed coarsely, and it is hard to learn from them and figure out how they could benefit future studies.

[Experiments]

The comparisons in Table 1 may not be fair enough. Some pre-training methods use different training steps and are based on different sizes of training data. It would be better to add some columns to indicate the training steps, data size, or training cost for more clear comparisons.

[Analysis]

The design and usefulness of the simple synthetic tasks lack theoretical analysis. Why are the simple tasks, e.g., Set and Identity, intuitively better than the existing methods (which may be more linguistically motivated, e.g., Nesting Dependency Artificial Language). How could the model learn from the identity task by copying the input to the output? Is there any supportive reference for the knowledge acquisition process?


[Detailed Concerns]

1.	This paper tries to explain the problem from trying large numbers of different synthetic pre-training settings. However, the performances of almost all of them are limited (i.e. worse than the original). From this perspective, the greatest contribution of this paper seems to be the two somewhat effective synthetic pre-training tasks (Set and Identify). In practical, besides, this paper does not present any definitive or helpful conclusions for future language pre-training. The claims in the paper are weak or somewhat unsure. The above drawbacks make the contribution of the paper weak.
2.	It shows that the LM can learn some simple clues from synthetic pre-training without being exposed to natural language. However, the gain on natural language in very limited against on artificial language. From Table 1, the performances on SQuAD are significantly dropped compared to pre-trained T5 (e.g. T5 77.5/86.0 -> LIME 50.4/62.1).
3.	Considering the great difference, it is possible that synthetic pre-training provides almost no gain for natural language learning. Further experiments are supposed to be done on natural language tasks (e.g. GLUE or GLUE sub-tasks).
4.	There seems to be an attempt to avoid the issue, making most experiments on a summarization task (CNNDM). The clues for summarization can be more easily to learn, rather than for reading comprehension.
5.	It shows that only keeping the pre-trained statistics can retain most pre-training benefit from synthetic pre-training. However, it needs to be justified that the similar situation (e.g. initialize the model with T5 pre-trained weights using the same methods) does not exist on natural language pre-training. Pre-training is supposed to learn the coarse distribution of a certain language.
6.	The pre-training settings is not comprehensive. Is one million data sufficient for synthetic pre-training? If with more pre-training data, the conclusions in the paper can be more credible. Another concern is that if the claims made in the paper is general to a bigger model. It is better to run the experiments on a bigger setting (e.g. 12 encoder layer + 12 decoder layer).
7.	What if we combine some of the great synthetic tasks together?

Minor Issues:

Use \citet{} for inner-text citations.
Line 63: no need to capitalize “nesting dependency”.
Line 91: what are those “original papers”? References needed.
Line 166: There is no Figure 4 in the draft.
Table 2: There is an unnecessary empty line in the table.

Typo?
Figure 4 tends to be Figure 1? I cannot find Figure 4 in the main paper.

---

> ### Author Response · Authors · 2022-08-02
> **Response to Reviewer 4 (Part 2 of 2)**
>
> **"From Table 1, the performances on SQuAD are significantly dropped compared to pre-trained T5 (e.g. T5 77.5/86.0 -> LIME 50.4/62.1). Considering the great difference, it is possible that synthetic pre-training provides almost no gain for natural language learning. Further experiments are supposed to be done on natural language tasks (e.g. GLUE or GLUE sub-tasks)."**
>
> On the one hand, we agree it is true for some natural language tasks synthetic pre-training provides less gain compared to natural pre-training. Past works (Chiang and Li (AAAI 2021)) also had similar observations where they found synthetic pre-training was significantly worse than natural pre-training on GLUE. On the other hand, we should acknowledge big gains from synthetic pre-training compared to no pre-training at all (e.g. LIME 50.4/62.1 vs. random init 17.2/25.2 on SQuAD).
>
> The goal of the paper is to identify and decompose the benefits of natural pre-training, rather than to match performances of natural pre-training with synthetic pre-training. In our discussion section, we remark that it is expected for synthetic pre-training to lag behind natural pre-training especially for tasks involving natural language. The surprising insight that we try to highlight with our experiments is how much gains we can obtain by doing full synthetic pre-training compared to no pre-training at all.
>
>
> **"[The paper] shows that only keeping the pre-trained statistics can retain most pre-training benefit from synthetic pre-training. However, it needs to be justified that the similar situation (e.g. initialize the model with T5 pre-trained weights using the same methods) does not exist on natural language pre-training."**
>
> With the statistics initialization experiments, we only intend to show the claim: "For effective gains on some downstream tasks, it is not necessary for the pre-training to have resulted in the model learning to do the pre-train task." Such a claim is True if we find that there exists even one pre-training task where initialization with statistics produces effective gains. Thus, the current experiment results we have are sufficient for proving it regardless of what the results are for T5 natural pre-training.
>
> However, the suggested experiment is definitely interesting to do, and we ran it in our preliminary studies. We observed that one does not have much gains from using pre-trained statistics of T5 natural pre-training. Since we do not find this experiment relevant to the main claim we wanted to emphasize, we did not put it in the paper. If the reviewer thinks it is still interesting to show, we can add this to the appendix in the future work.
>
> **What if we combine some of the great synthetic tasks together?**
>
> In our preliminary studies, we ran some experiments where we pre-trained on multiple synthetic tasks together (LIME together with all 18 simpler synthetic tasks) and found there are slight improvements over just training solely on LIME (on CNN-DM it got 35.2).
> Since the motivation of the paper is to understand pre-training by simplifying it to simple synthetic pre-training. We do not think combining many synthetic tasks is relevant to the main claim we want to emphasize. On the contrary, it actually may complicate our understanding. Hence we did not put it in the paper. But we agree with the reviewer that this is an interesting direction to explore if one wants to further empower synthetic pre-training.
>
> **"Why are the simple tasks, e.g., Set and Identity, intuitively better than the existing methods (which may be more linguistically motivated, e.g., Nesting Dependency Artificial Language). How could the model learn from the identity task by copying the input to the output? Is there any supportive reference for the knowledge acquisition process?"**
>
> See the second section in the common response labeled "Lack of explanation why Set and Identity help."

---

> ### Author Response · Authors · 2022-08-02
> **Response to Reviewer 4 (Part 1 of 2)**
>
> **"[This] paper does not give a clear answer to the core question raised in the introduction, 'What properties of pre-training are sufficient for effective gains.'"**
>
> Our original phrasing of the core question was inaccurate. In our new version of the paper with revisions for better organization and clarity, we have changed it to: "What properties of pre-training are necessary for effective gains?"
> Our paper's three sets of simplifying experiments help understand this question by showing what properties are not necessary for effective gains.
> - For effective gains on some downstream tasks, it is not necessary for the pre-training to involve natural data.
>    - LIME, a fully synthetic task, provided effective gains across all downstream tasks.
> - For effective gains on some downstream tasks, it is not necessary for the pre-training to involve a complex task designed to fit the downstream task structure.
>     - Set performance matched the best synthetic task LIME and was better than the other two synthetic tasks despite having much simpler design: it does not represent three fundamental logical operations as LIME does, it does not having a structural property that mimics natural language as the Artificial Language task does, and it does not reflect operations used for summarization as the Nonsense Summary task does.
> - For effective gains on some downstream tasks, it is not necessary for the pre-training to have resulted in the model learning to do the pre-training task.
>     - The models initialized via synthetic pre-trained statistics would be unable to do the synthetic task, yet still get significant downstream task benefit.
>
> **"The motivation is not clear. I would suggest motivating this work by discussing some limitations of the current synthetic pre-training methods. According to the discussions in Sections 4-5, existing synthetic tasks might have disadvantages, e.g., computation complexity/cost, so they are not optimal as the pre-training tasks, and more generic ones (proposed in this work) would be better."**
>
> The main motivation of the work is helping understand the core question, "What properties of pre-training are necessary for effective gains?" The motivation for discovering Set and Identity is not that they are more practically useful than the existing synthetic pre-training, but rather that their existence helps with understanding the core question, "What properties of pre-training are necessary for effective gains?" Set and Identity show that it is not necessary for effective pre-training to involve a complex task designed to fit the downstream task structure, which the existing synthetic pre-training have. Also, Set and Identity provide a simple starting point for future investigations towards explaining why pre-training helps. Fully understanding why such simple tasks produce benefits is potentially easier than fully understanding why something complex like masked language modeling produces benefits.
>
> **"In practical, besides, this paper does not present any definitive or helpful conclusions for future language pre-training. The claims in the paper are weak or somewhat unsure. The above drawbacks make the contribution of the paper weak."**
>
> As stated above, the paper's contribution is not to prescribe any immediate improvements to training transformers, but rather providing insights that help with understanding. These insights reveal future directions of research that may lead to practical benefits. For example, the paper demonstrates the importance of the scale of weight initialization, which future work can build on to design better weight initialization schemes for training transformers.
>
> **"The comparisons in Table 1 may not be fair enough. Some pre-training methods use different training steps and are based on different sizes of training data… The pre-training settings is not comprehensive. Is one million data sufficient for synthetic pre-training?"**
>
> We state in the paper that for all synthetic tasks, we used training data size of 1 million. For all synthetic tasks besides the Artificial Language task, we pre-trained until the first checkpoint that got greater than 99% validation accuracy on the pre-train task. For the Artificial Language task which could not reach 99% validation, we took the checkpoint training step where max validation accuracy occurred and verified that doubling the training data set size to 2 million did not improve the maximum validation accuracy on the pre-training task and also did not improve performance on the downstream tasks.
> In essence, for each task we chose training steps and training data size sufficient for maximizing validation accuracy. If validation accuracy is a reliable indicator of the effectiveness of pre-training with a particular task, then our selection of training data size and training steps is fair, since for each task we attain the maximum validation accuracy achievable for that task even if unlimited training data and steps were allowed.

---

> > ### Comment · Reviewer_6Qmz · 2022-08-08
> > **check later**
> >
> > Thanks for the feedback, I would like to make an apology for the later reply as I have been on a trip since last month. I promise I will read all feedbacks before the ddl and take necessary actions by considering the authors feedbacks if I cannot make a prompt enough reply during the discussion.
> >
> > As much more materials were added in the rebuttal stage, I will need time to read them carefully. My official response for this updated version will given later.

---

> ### Author Response · Authors · 2022-08-04
> **Follow-up to Response**
>
> Dear reviewer,
>
> We would like to follow up on our response. Please let us know if we have addressed your concerns or if you have further questions. We hope to address them as much as we can. Thank you again for your feedback.
>
> Looking forward,
> Authors of Paper 6848

---

### Official Review · Reviewer_iZRj · 2022-07-11

**Rating:** 7
**Confidence:** 4
**Soundness:** 3 good
**Presentation:** 4 excellent
**Contribution:** 3 good

**Summary:**

This paper investigates what properties of synthetic pretraining lead to better performance on downstream NLP tasks. The authors find that very simple synthetic tasks perform at a comparable level to prior state-of-the-art methods. For example, a straightforward identity task where the model must output the same tokens provided as input. This motivates further steps to simplify, where only the use of weight initialization statistics is sufficient to achieve much of the gains of pretraining.


**Questions:**

- Maybe I missed the details, but how is the random init baseline trained? Is poor weight initialization not something that could be overcome with sufficiently long training?


**Limitations:**

The authors do a good job acknowledging that this investigation only scratches the surface of a rich and complex area and there's much more to be done in understanding the benefits of pretraining.

**Strengths And Weaknesses:**

Strengths:

- This is a solid paper, it is written well and provides an interesting investigation uncovering insights that could otherwise be easily overlooked in the community.
- I appreciate how the paper is organized and the journey we go on of further simplification. The process of stripping away complexity piece by piece until we are down to just a hyperparameter for initializing layer norm is in my opinion much more insightful than if the authors had instead chosen to present some new elaborate task that bumped performance by half a percent.
- The experiments follow a natural trajectory and are fairly comprehensive.

Weaknesses:

- The insight into initialization of layer norm parameters is interesting, and not something that I imagine most people would pay attention to without this sort of investigation. One question is whether knowing that this is an important parameter concretely changes the results one can achieve in these pretraining/finetuning regimes. Have the authors seen any net improvement to performance (higher accuracies, shorter training times, etc) by going back and pretraining and finetuning but this time with a better initialized model?
- One part that nags at me is that most of the comparisons of the paper report relative performance along a linear scale (e.g. method A closed 70% of the gap between random init and natural language pretraining). On one hand, it’s not necessarily an inappropriate  way of presenting numbers, but on the other, further improvements to performance are much more difficult than those initial gains. It could be trivial to get most of the way to matching a particular baseline (and in some ways this is shown with the weight initialization results) and yet it might be impossible to close the rest of the gap. This isn’t reflected in the way results are reported and discussed throughout the paper - the reader doesn’t necessarily have any way of knowing how much more difficult it is to get to say, 35.8 from 32.8 even though that certainly seems very close from a baseline of 18.9. And with the random init baseline being so poor, all of the relative numbers look better for it.
- It would have been interesting to see some comments on the relative performance of the various simple synthetic tasks. I found it fascinating that there was a clear distinction of the worst performing tasks being the only tasks where the output did not involve reusing input tokens (count, length, search, longest word). I know there’s only so much real estate for discussion, but it’s not clear why some simple tasks would fare so well while others so poorly, so some discussion along these lines might make sense to include.

---

> ### Author Response · Authors · 2022-08-02
> **Response to Reviewer 3**
>
> **"Maybe I missed the details, but how is the random init baseline trained?"**
>
> For the random init baseline, there is no pre-training done. The T5 model is initialized according to the default way T5 is randomly initialized (e.g. fully connected layers via Xavier Initialization).
>
> **"Have the authors seen any net improvement to performance (higher accuracies, shorter training times, etc) by going back and pretraining and finetuning but this time with a better initialized model?"**
>
> See the third section in the common response labeled "Combining synthetic pre-training and natural pre-training."

---

> ### Author Response · Authors · 2022-08-04
> **Follow-up to Response**
>
> Dear reviewer,
>
> We would like to follow up on our response. Please let us know if we have addressed your concerns or if you have further questions. We hope to address them as much as we can. Thank you again for your feedback.
>
> Looking forward,
> Authors of Paper 6848

---

### Official Review · Reviewer_9nVq · 2022-07-11

**Rating:** 6
**Confidence:** 3
**Soundness:** 3 good
**Presentation:** 3 good
**Contribution:** 3 good

**Summary:**

This paper provides a series of empirical studies to find the key components in synthetic pre-training for language tasks. Surprisingly, it reveals that two simple synthetic tasks: 'set' and 'identity', can achieve similar performance compared with more sophisticated ones. It also shows that using the statistics of the pretraining weight to initialize the downstream model can also result in similar performance.

**Questions:**

1. More insights on the success of the 'set' and 'identity' function.
2. Clarity of the discussions in Line 197 to 200.

Please refer to the previous section for more details.

**Limitations:**

I do not see issues in this aspect.

**Strengths And Weaknesses:**

### Strengths

1. The experimental analysis is designed coherently and sufficiently. It enumerates a series of simple synthetic tasks and finds two surprisingly simple but effective strategies: set and identity. It also studies how to improve the initialization strategy using only the statistics of the pre-training weight. All these findings may provide future research with ideas on pre-training as the title suggested.
2. The overall writing is coherent and the proposals are easy to follow.

### Weaknesses

1. I am a little worried about the technical novelty. On the one hand, all the findings are based on experimental studies without specific designs. For example, it is hard to tell why 'set' and 'identity' functions are more suitable than others. Missing such explanations may result in limited technical insight. On the other hand, the focus on the normalization layer in Sec. 5.3 cannot be viewed as brand-new. For example, a lot of studies have revealed that training only the parameters of batch normalization layers can also result in an ImageNet classifier with reasonable accuracy.
2. Some discussions are confusing to me. For example, in Line 197 to 200:
> We note that it does not close the possibility that better initialization statistics is the sole cause of benefits because there exist other initialization schemes such that the statistics of the pre-trained model are a subset of the statistics from the initialization scheme.

    I could not get the connection between the reason and the result.

Despite some concerns mentioned above, I find that the overall contribution of this paper is strong enough to bring insights on synthetic pretraining into the research community. Thus, I vote for 'weak accept' temporarily. Discussions on the above points are warmly welcome.

---

> ### Author Response · Authors · 2022-08-02
> **Response to Reviewer 2**
>
> **"Clarity of the discussions in Line 197 to 200."**
>
> Consider the statement: better initialization statistics is the sole cause of benefits of LIME, Set, and Identity pre-training.
>
> Towards exploring this statement, we compared the performance between initializing with the mean and SD of each synthetic pre-trained parameter (Per_Param_Mean_SD) and initializing with the exact weights of synthetic pre-trained parameters as normally done when fine-tuning (Full_Init).
>
> If the performance were equal, then the statement would be True.
>
> We found that the performance was not equal: Per_Param_Mean_SD was clearly worse than Full_Init.
>
> With this result, we have not shown that the statement is True. However, this result does not imply that the statement is False: there exist other initialization schemes using weight statistics besides Per_Param_Mean_SD that we did not try and which might be able to attain equal performance.
>
> **"More insights on the success of the 'Set' and 'Identity' function."**
>
> See the second section in the common response labeled "Lack of explanation why Set and Identity help."

---

> ### Author Response · Authors · 2022-08-04
> **Follow-up to Response**
>
> Dear reviewer,
>
> We would like to follow up on our response. Please let us know if we have addressed your concerns or if you have further questions. We hope to address them as much as we can. Thank you again for your feedback.
>
> Looking forward,
> Authors of Paper 6848

---

### Official Review · Reviewer_tNV7 · 2022-07-13

**Rating:** 5
**Confidence:** 3
**Soundness:** 2 fair
**Presentation:** 3 good
**Contribution:** 3 good

**Summary:**

The authors explore synthetic pretraining of T5 style language models. They investigate previously proposed synthetic tasks and introduce simpler synthetic tasks that they find to provide similar benefit to the more complicated previously proposed synthetic tasks.

**Questions:**

As you find that synthetic pretraining provides you with a model performing "in between" of random initalization and real pretrained model, have you considered taking the synthetically pretrained model as initialization for real pretraining? If so, are you able to reduce the number of pretraining steps on real-world data while achieving same or better quality?

**Limitations:**

The authors only evaluate on T5-Small with 60M parameters. It remains unclear to what degree synthetic pretraining would help on larger models. The authors also do not investigate to what degree synthetically pretrained models can be continually pretrained on real world data and if such continued pretraining can close the gap in performance to the pretrained T5 with less real world data than what the T5 model saw during pretraining.

**Strengths And Weaknesses:**

The authors find that simple synthetic tasks can perform similar to synthetic tasks from the literature, which is original and interesting. Unfortunately the authors do not touch on key questions opened up by their results, such as investigating if results hold on larger models (ie. if the percentage of the gap between Random Init and Pre-trained T5 closed by synthetic pretraining remains constant) or if continued real pretraining with the initialization obtained from synthetic pretraining can reduce the amount of data required for real pretraining. Thus the work appears to be in a preliminary, though encouraging state that is relevant to inform further research.

---

> ### Author Response · Authors · 2022-08-02
> **Response to Reviewer 1**
>
> **"The authors only evaluate on T5-Small with 60M parameters. It remains unclear to what degree synthetic pretraining would help on larger models."**
>
> Thank the reviewer for the suggestion. We have run the experiments for a large model, and we found similar observations. See the first response in the common response.
>
>
> **"The authors also do not investigate to what degree synthetically pretrained models can be continually pretrained on real world data and if such continued pretraining can close the gap in performance to the pretrained T5 with less real world data than what the T5 model saw during pretraining."**
>
> Thank the reviewer for the a good question. See the last entry in the common response labelled "Combining synthetic pre-training and natural pre-training."

---

> ### Author Response · Authors · 2022-08-04
> **Follow-up to Response**
>
> Dear reviewer,
>
> We would like to follow up on our response. Please let us know if we have addressed your concerns or if you have further questions. We hope to address them as much as we can. Thank you again for your feedback.
>
> Looking forward,
> Authors of Paper 6848

---

### Author Response · Authors · 2022-08-02
**Common Response**

We thank all reviewers for their thoughtful and constructive reviews. In this common response, we address the concerns shared by multiple reviewers, before we answer individual reviewer's questions in separate posts.


**Results on larger model**

Reviewers 1 and 4 mention that it is unclear whether results hold on larger models. We ran new experiments with T5 Base, which has 220 million total parameters with 12 encoder/decoder layers compared to 60 million total parameters with 6 encoder/decoder layers for T5 Small.  LIME, Set, LIME Per_Param_Mean_SD, and Set Per_Param_Mean_SD close a large proportion of the gap to natural pre-training, similar to what happens with T5 Small. We post results in Appendix J, Table 9.

|    T5-Base Exps    |                       |                  |                   |                         |             |        |
|:----------------------:|:---------------------:|:----------------:|:-----------------:|:-----------------------:|:-----------:|:------:|
|                        | CNNDM10K (ROUGE1) | MTOP (EM/F1) | SQuAD (EM/F1) | Retrosynthesis (EM) | Average |        |
|      Pretrained_T5     |          37.6         |     83.8/96.8    |     83.6/91.1     |           44.4          |     62.4    | 100.0% |
|       Random_Init      |          16.4         |    30.30/78.3    |      7.5/9.9      |           36.9          |     22.8    |  0.0%  |
|          LIME          |          33.7         |     72.5/93.8    |     48.4/60.4     |           38.2          |     48.2    |  57.9% |
|           Set          |          34.4         |     74.4/94.2    |     51.8/62.7     |           39.6          |     50.1    |  65.4% |
| LIME Per_Param_Mean_SD |          26.5         |     59.2/90.5    |     25.9/34.2     |           37.3          |     37.2    |  32.8% |
|  Set Per_Param_Mean_SD |          27.4         |     64.9/92.2    |     22.4/30.6     |           37.5          |     38.1    |  36.0% |


**Lack of explanation why Set and Identity help**

Reviewers 2 and 4 mention that there is a lack of explanation why Set and Identity help.

First, we try to make some progress towards explaining why Set and Identity help through our section on exploring benefits from better initialization statistics. We find that a large amount of benefits from synthetic pretraining is derived from better initialization statistics. It also seems from our empirical results that synthetic pre-training with tasks like Set and Identity can already provide good initialization statistics. We leave more in-depth explanations of the latter for future work..

Second, we note that our main intended contribution in this paper is identifying that Set and Identity provide effective pre-training gains. The effectiveness of Set and Identity shows that it is not necessary for effective pre-training to involve a complex task designed to fit downstream task structure. Also, Set and Identity provide a simple starting point for future investigations towards explaining why pre-training helps. Fully understanding why such simple tasks produce benefits is potentially easier than fully understanding why something complex like masked language modeling produces benefits. In future work, we will try doing exploration and analysis on exactly why Set and Identity help.


**Combining synthetic pre-training and natural pre-training**

Reviewers 1 and 3 ask what may happen if we do synthetic pre-training before natural pre-training. In our preliminary studies, the most naive strategy of combining them gave some negative results: we tried first doing LIME synthetic pre-training before 10K steps of natural pre-training, and we found that this produced no difference in CNNDM-10K, MTOP, and SQuAD performance compared to only doing 10K steps of natural pre-training. We believe that further innovation is needed for seeing the benefits of combining, and we leave those studies to future work.

---

### Meta-Review · Area_Chair_vCSG · 2022-08-28

**Recommendation:** Accept
**Confidence:** Less certain

**Metareview:**

# Overview

This paper presents a fascinating research question: "What properties of pre-training are necessary for effective gains" in natural language processing settings? The paper also tackles it in a fascinating way: exploring several synthetic tasks in exacting detail. This is an empirical paper whose goal is to consolidate existing knowledge on the topic of synthetic pre-training and pose a scientific question about the nature of pre-training.

What does it look like for such a paper to be publishable? There are several things it _doesn't_ need to do. That includes:
* Setting a new SOTA for anything, synthetic pre-training or otherwise.
* Providing a theoretical account for empirical phenomena.
* Completely or definitively answer the provided research question, which is a massive undertaking that will require an entire research literature, not a single paper.

However, what such a paper does need to do is to be absolutely rigorous in the situations that it purports to address and to follow empirical best practices. That includes:
* Considering a comprehensive range of downstream tasks (that provides a basis for comparison between methods) and considering them across the full range of experiments.
* Reporting all data clearly in ways that fairly state what was discovered, even if some results are mixed or negative.
* Running relevant ablations to dig into the core research question.
* Consider new approaches to the problem inspired by those findings to test the predictive power of those findings.

# Reviews

## Reviewer 6Qmz

This reviewer raises a number of concerns about the paper, some of which I have discarded based on the criteria above. For example, I have ignored the following concerns since they set an unfair bar for a scientific paper taking on a vast question:
* _This paper does not give a clear answer to the core question raised in the introduction._
* _The design and usefulness of the simple synthetic tasks lack theoretical analysis._
* _The performances of almost all of the[ synthetic tasks] are limited (i.e., worse than the original.)_
* _This paper does not present any definitive or helpful conclusions for the future of language pre-training._

However, the reviewer raises several important methodological concerns that undermine confidence in the empirical findings of the paper:
* _The gain on natural language is very limited against on artificial language._ _The performance on SQuAD are significantly dropped._ (AC note: In general, I found describing the data as "recovering XX% of the change in accuracy from pre-training T5" to be a pretty misleading way to exaggerate relatively small improvements of a handful of percentage points over random initialization. If the results are already strong, there's no need to do this. It's not a reason to reject, but it made me fear that there were other things hidden in the paper that I didn't find. The big question for this reviewer seems to be "are these results good enough to suggest that something interesting has been found via these experiments?" It's in the eye of the beholder, and that's part of the challenge of getting in a paper like this.)
* _There seems to be an attempt to avoid the issue, making most experiments on a summarization task._
* The reviewer also suggests some fantastic follow-on experiments that would be great ablations or ways of evaluating the predictive power of the proposed findings.

In general, this reviewer questions the rigor of the evaluation of the pre-training methods studied. That's of paramount concern for a paper like this.

## Reviewer iZRj

This reviewer had several questions that were addressed in the author response. One question that remains unaddressed:
* _One part that nags at me is that most of the comparisons of the paper report relative performance along a linear scale._ (The AC shares this concern and found this frustrating. It overstates the quality of the results, since each marginal improvement in performance is difficult to accomplish. SQuAD scores are still dozens of percentage points worse than T5, for example. I think this may mislead readers - and possibly misled some of the reviewers - into thinking the results were better than they actually were.)

## Reviewer 9nVq

This reviewer had a couple of concerns. Most prominently:
* _Missing such explanations [for why these simpler methods work] may result in limited technical insight._ (I agree, but it's a tall order to do this definitively and that's an unfair bar for acceptance.)

## Reviewer tNV7

This reviewer was primarily concerned with the scale of the experiments. I also share this concern. Even a 200M parameter T5 model is relatively small all things considered. (I understand that the authors may not have enough compute available to go after bigger models with this range of pre-training and evaluation techniques, but - if that's a concern - than the overall topic of understanding T5 pre-training may not be the right problem for them to go after.) Analyzing how these techniques do as scale increases is an important aspect of the question the authors are taking on. Do these synthetic tasks top out at some point, while pretraining on natural language data continues to help? Does the gap keep increasing? These are big questions in a field that is completely preoccupied with all things "scale" right now, so it's a big missing piece of this paper in my view.

# Recommendation: Accept

I argue in favor of accepting this paper. Despite the range of flaws pointed out by the reviewers and this AC, this paper takes a productive scientific step toward understanding the properties of pre-training tasks that make pre-training effective. This paper is a great survey of the current state of the art and fills in many gaps in our knowledge about how successful pre-training is at various levels of complexity. Yes, there is much more work to do, and I hope these authors and other members of the community build on this work to do so.

# Please please PLEASE fix a few things!

With that said, there are a few things that I beg and implore the authors to do in the camera ready (in order of priority for me). Without doing these things, this paper may not serve as a solid enough foundation for future work that people can readily build on it, and the paper's impact will be dramatically less than it could be (scientifically and in terms of citations). This is for your own good!

1. Ensure that all experiments are evaluated on all fine-tuning tasks. At least provide that data in appendices. I know you were frustrated with Reviewer 6Qmz, but the reviewer was completely right on this and several other points.
2. Get rid of those annoying comparisons made about "percentage of gap closed." Just talk about percentage points on the actual task. It's confusing and misleading, and it overstates the efficacy of these methods. It doesn't matter if these methods are good or not: the important part is filling in gaps in our scientific knowledge. This bad way of describing the data isn't _quite_ enough to reject over on its own, but it seems to have annoyed the heck out of several of the reviewers and it _really_ annoyed the heck out of me.
3. Add the T5 200M results to the paper and add a discussion of how these results change as you scale up. Ideally, add one larger scale so you have three points and can start to see trends. This is a place where the "percentage of gap closed" metrics make more sense as a way to compare across scales.
4. Add GLUE. The world of NLP fine-tuning tasks is vast, and everyone has their favorites. Make sure everyone's favorite is there so nobody can complain.

# Notes to Authors

All of the reviewers agreed that this was an interesting scientific question, and I encourage the authors to continue building on this line of work. In addition, all of the reviewers did respond to the rebuttal, although several appear to have provided their private thoughts to me rather than broadcasting it (including Reviewer 6Qmz). I have taken those thoughts into account when putting together this metareview.


**Award:**

No

---

### Decision · Program_Chairs · 2022-09-14

Accept